# Viral Hepatitis A, B and C in a Group of Transgender Women in Central Brazil

**DOI:** 10.3390/tropicalmed7100269

**Published:** 2022-09-27

**Authors:** Lucila Pessuti Ferri, Priscilla dos Santos Junqueira, Mayara Maria Souza de Almeida, Mariana Gomes Oliveira, Brunna Rodrigues de Oliveira, Bruno Vinícius Diniz e Silva, Larissa Silva Magalhães, Lívia Melo Villar, Karlla Antonieta Amorim Caetano, Márcia Maria Souza, Megmar Aparecida dos Santos Carneiro, Regina Maria Bringel Martins, Sheila Araujo Teles

**Affiliations:** 1Faculty of Nursing, Federal University of Goiás, Goiânia 74690-900, Brazil; 2Institute of Tropical Pathology and Public Health, Federal University of Goiás, Goiânia 74690-900, Brazil; 3Laboratory of Viral Hepatitis, Oswaldo Cruz Institute, FIOCRUZ, Rio de Janeiro 21040-900, Brazil

**Keywords:** hepatitis A, hepatitis B, hepatitis C, immunogenicity, transgender women, vaccination, viral hepatitis

## Abstract

Transgender women (TGW) have limited access to affordable viral hepatitis testing, hepatitis B vaccination, and treatment. We aimed to estimate the prevalence of viral hepatitis A, B, and C, as well as to compare the adherence and immunogenicity of two hepatitis B vaccine schedules among TGW in Central Brazil. A total of 440 TGW were interviewed and tested for hepatitis A, B, and C serological markers from 2017 to 2018. The hepatitis B vaccine was offered to 230 eligible TGW: 112 received a super accelerated hepatitis B vaccine schedule (G1) and 118 a standard schedule (G2). The antibody against the hepatitis A virus (HAV) was detected in 75.63% of the participants, and 12.3% of the TGW were exposed to the hepatitis B virus (HBV). Two (0.46%) participants were HBV carriers. Only 41.5% of the participants showed a serological profile of hepatitis B vaccination. The antibody against the hepatitis C virus (anti-HCV) was found in six participants (1.37%). Of the TGW who received the first vaccine dose, 62 (55.36%) and 49 (41.52%) in G1 and G2, respectively, received three doses (*p* = 0.036). The vaccine response was evaluated in 28 G1 and 22 G2 TGW; of these, 89.3% and 100% developed protective anti-hepatitis B surface-antigen titers, respectively (*p* = 0.113). Since one-third of younger transgender women are susceptible to HAV, hepatitis B immunization is low, and the anti-HCV rate is higher in this group than in the general population in Central Brazil, public-health attention is warranted. The super-accelerated scheme demonstrated better adhesion and good immunogenicity, suggesting that it would be a more cost-effective solution.

## 1. Introduction

Viral hepatitis is characterized by high morbidity and mortality worldwide, with 325 million carriers of hepatitis B and C estimated in 2015 to be at risk of progression to cirrhosis, hepatocellular carcinoma, and death [1].

Hepatitis B virus (HBV) is easily transmitted by sexual, parenteral, and vertical routes [2]. Meanwhile, the hepatitis C virus (HCV) is disseminated predominantly by the parenteral route. However, sexual transmission has been reported, especially among people with human immunodeficiency virus (HIV) who practice unprotected anal sex [2,3,4].

Hepatitis A virus (HAV) infection is an acute self-limiting hepatitis. However, this infection may be a significant cause of hospitalization and absenteeism. HAV is transmitted primarily by the fecal–oral route through contaminated water or food [2]. In addition, HAV transmission through oro-anal sex has been reported among men who have sex with men (MSM) [2,5].

Hepatitis A and B are vaccine-preventable diseases. In Brazil, infant vaccination for hepatitis B has been offered since the early 2000s and gradually extended to the general population. The hepatitis A vaccine has been recommended for infants since 2014, following the change in Brazilian hepatitis A endemicity from high to intermediate [6]. There is no vaccine against hepatitis C, but considerable advances have been achieved in its treatment. Currently, acute and chronic hepatitis C may be treated and cured [4].

Transgender people have a gender identity or gender expression that is different from their assigned birth sex [7]. Transgender women (TGW) are disproportionately affected by sexually transmitted infections due to frequent unsafe sex, illicit drug consumption, and reduced access to health services [8,9]. However, there are only a few studies on hepatitis B and C among TGW [10]. Additionally, when some studies group TGW as men who have sex with men, they minimize the differences and diversities among these highly disparate populations (MSM) [11,12,13].

The screening of high-risk population subgroups is essential to achieving the global health goal of eliminating viral hepatitis as a public health problem by 2030 [14]. However, most socially vulnerable people (such as TGW) have limited access to affordable hepatitis testing, hepatitis B vaccination, and treatment. Further, the long period between the second and third vaccine doses of the conventional hepatitis B vaccine schedule has compromised compliance with the full-vaccine scheme. Therefore, some authors have recommended hepatitis B vaccine schemes with shorter intervals between doses for hard-to-reach populations [15,16,17], such as TGW.

The Brazilian health-surveillance system does not collect data on gender identity. Studies on viral hepatitis among TGW are scarce, and there are no data on hepatitis B vaccination. Therefore, this study aimed to estimate the prevalence of hepatitis A, B, and C among TGW in Central Brazil. In addition, we compared the adherence and immunogenicity of two hepatitis B vaccine schedules to contribute to public health for this socially marginalized population.

## 2. Materials and Methods

The respondent-driven sampling method (RDS) was used to recruit TGW in three cities in Goiás, Central Brazil, between April 2018 and August 2019. This sampling method is used for populations of research subgroups that are difficult to access [18]. Eight key TGW, denominated as “seeds”, with diverse sexual identities, age, education, and jobs started the recruitment chains. These women were selected based on their high interest in participating in the study, their self-reporting of expressive social networks, and their high popularity in the TGW community. Each seed received three coupons for referral and one reward coupon. Participants recruited by seeds who enrolled in the study were given three coupons for further recruitment. Persons who self-defined as transgender women and presented a valid RDS coupon were included in the study. Persons who were found to be noticeably under the effects of psychoactive drugs (drunk or incoherent) were excluded.

The minimum required sample, with a significance level of 95% (<0.05), a precision of 3.0%, a design effect of 2.5, and a prevalence for anti-HBc at 12.7% [19], was 426 TGW.

Initially, all 440 participants were informed of the objectives and methodology of the study and signed written informed consent. The ethics committee waived the written consent of minors’ parents and guardians. Participants were then interviewed face-to-face on sociodemographic characteristics, sexual behaviors, consumption of alcohol and illicit drugs, and previous hepatitis B vaccination.

### 2.1. Viral Hepatitis Serological Markers

A total of 439 participants also agreed to provide a blood sample, and their samples were tested by enzyme-linked immunosorbent assay (ELISA) for antibodies against hepatitis A (total anti-HAV) (Bioelisa, Biokit, Barcelona, Spain); hepatitis B surface antigen (HBsAg), anti-HBs and anti-hepatitis B core antigen (HBc) total, and anti-HCV (Bioelisa-Bioclin^®^, Quibasa, Belo Horizonte, Brazil) markers. Samples that were total anti-HAV positive were retested for detection of IgM anti-HAV (Bioelisa, Biokit, Barcelona, Spain).

To compare the response to HBV vaccine in TGW who received a super-accelerated hepatitis B vaccine schedule (G1) and a standard schedule (G2), blood samples (3 mL) were collected approximately 45 to 60 days after the third (G1 and G2) and fourth vaccine dose (G1), and tested for quantitative anti-HBs by chemiluminescence. Anti-HB titers of ≥10 mIU/mL were considered protective.

### 2.2. Hepatitis B Vaccination

For logistic convenience, the analysis of two hepatitis B vaccine schemes was performed only in Goiania. The study included 230 TGW who reported no previous hepatitis B or were unaware of their hepatitis B vaccination status in a hepatitis B vaccination cohort. They were randomly recruited to receive either a super -accelerated scheme (G1; four doses at 0, 7, 21, and 180 days) or a standard scheme (G2; three doses at 0, 1, and 4 months) (Figure 1). Vaccine doses of 20 µg of recombinant HBsAg were administered into the deltoid muscle (Serum Institute of India PVT. LTD; lots 03560L24 and 03560L72).

Vaccine doses and blood draws were scheduled by cell phone, and the participants chose the place, time, and day (public health facilities or homes). For each appointment, three calls were made, and three WhatsApp (Menlo Park, CA, USA) messages were sent. If the participant did not respond after all contact attempts, they were considered to have dropped out.

### 2.3. Data Analysis

Data were analyzed using the STATA statistical software v.13 SE (College Station, TX, USA). The prevalence of viral serological markers and geometric mean titers (GMTs) of anti-HBs antibodies were calculated with a 95% confidence interval (CI). The Chi-squared and Fisher’s exact tests were used to evaluate differences between proportions. To identify variables associated with HBV exposure, TGW who showed positivity for anti-HBs alone (previous hepatitis B vaccine) were excluded from the analysis. The outcome of HBV exposure was defined as positivity for the anti-HBc marker. Variables with *p* < 0.20 were included in a stepwise logistic regression model. Statistical significance was defined at the 0.05 probability level in all analyses.

This study was approved by the Ethics Committee for Human Research of the Universidade Federal de Goiás (protocol number CAAE 77481417.5.0000.5083 and protocol number 2.358.818). All TGW who tested HBsAg positive or anti-HCV positive were referred to public health care services for medical follow-up and, if necessary, treatment.

## 3. Results

### 3.1. Characteristics of Transgender Women

A total of 440 TGW participated in the study, of whom 285 were recruited in the metropolitan region of Goiânia and 155 in inner cities in Goiás. They were predominantly young adults (median age, 25 years (16–59; interquartile range [IQR], 9)), single/divorced/widow, non-white, with low education and low income (median monthly income: R$2000/U$512; IQR, R$2000/U$512). Approximately only one-third of TGW reported condom use during their last sexual intercourse, and 47.84% had oro-anal sex. The majority exchanged sex for money, drugs, and goods (89.32%) and practiced insertive anal sex (74.26%) and receptive anal sex (97.49%). Daily alcohol consumption and illicit injection drug use were reported by 13.33% and 2.74%, respectively, while non-injection illicit drug use was reported by 74%. Of the total, 17.5% had a previous positive HIV test, and 50% had a history of sexually transmitted infection (STI). Industrial silicone (for cosmetic purposes) was found in 44.9% of the TGW (Table 1).

### 3.2. Prevalence of Hepatitis A, B, and C Serological Markers

Total anti-HAV positivity was found in 332 (75.63%; 95% CI, 71.4–79.74) participants, indicating previous exposure to HAV. Of these, one woman aged 21 years had serological evidence of current HAV infection (anti-HAV IgM positive) (Table 2). She was aged 21 years and reported sex work, alcohol and drug use, and the non-use of condoms during oral sex. The positivity for total anti-HAV ranged from 64.03% (73/114; 95% CI, 54.9–72.25) among those aged ≤21 years to 85.44% (95% CI, 77.35–90.97) among those aged ≥30 years (*p* < 0.01).

Anti-HCV positivity was detected in samples of six TGW (1.37; 95% CI, 0.63–2.95) (Table 2) who were unaware of their HCV serostatus. They were aged 27 to 54 years and reported sexual or parenteral risk behaviors (five did not use a condom during their last sexual intercourse; five had a previous STI; four reported non-injection drug use; one reported injection drug use; and two reported clandestine application of industrial silicone). Four anti-HCV-positive TGW were living with HIV.

HBV-infection markers were detected in 12.3% (54/439; 95% CI, 9.55–15.7) of the participants (Table 2), ranging from 4.39% (5/114; 95 CI, 1.89–9.86) among those aged ≤21 years to 30.09% (31/103; 95% CI, 22.09–39.54) among those aged ≥30 years (*p*< 0.01). As shown in Table 2, 37 TGW (8.43%; 95% CI, 6.18–11.4) were anti-HBc- and anti-HB-positive, and 15 TGW were anti-HBc-positive only (3.42%; 95% CI, 2.08–5.56), indicating previous HBV exposure. Two TGW (0.46%; 95% CI, 0.12–1.65) were HBV carriers (HBsAg and anti-HBc positive), while anti-HB positivity was detected in 182 (41.46%; 95% CI: 36.94–46.12) TGW, indicating previous hepatitis B vaccination. The proportion of TGW who showed a serological profile of hepatitis B vaccination did not differ among the age groups (*p* = 0.421).

Age, monthly income, sex work, and previous positive HIV testing or unknown status (*p* < 0.20) were included in a multiple logistic regression model, and only TGW aged over 25 years remained statistically associated with HBV exposure (Table 3).

### 3.3. Adherence to Hepatitis B Vaccination and Immunogenicity

The first hepatitis B vaccine dose was administered to 230 TGW who were randomly assigned to receive an accelerated vaccine scheme (G1; *n* = 112) or a standard scheme (G2; *n* = 118) (Figure 1). There was no difference between the groups in terms of age, education, and color (*p* > 0.05). The second vaccine dose was administered to 80/112 (71.43%) and 84/118 (71.19%) (*p* = 0.968), and the third vaccine dose to 62/112 (55.36%) and 49/118 (41.53%) in G1 and G2 (*p* = 0.036), respectively. In G1, only 23 TGW received the fourth dose. The median times between the first and second doses were 8 days (IQR, 7.75) and 34.5 days (IQR, 22.5) and the median times between the second and third doses were 21 days (IQR, 16) and 124 days (IQR, 60) for G1 and G2, respectively. In G1, the interval between the third and fourth doses was 163 days (IQR, 37).

Blood samples were obtained from 28 G1 and 22 G2 TGW following the third vaccine dose and tested for quantitative anti-HBs. Protective anti-HBs titers were detected in 89.29% of TGW in G1 and 100% in G2 (*p* = 0.113). The geometric mean of titers was 293.91 UI/mL (95% CI, 142.74–605.21) and 721 UI/mL (95% CI, 524.71–990.70) for G1 and G2, respectively. In 13 G1 women who received the fourth vaccine dose, the geometric mean of titers of anti-HBs was 331.78 UI/mL (95% CI, 113.07–973.55). In G1, three women did not respond with anti-HB protective titers following the third dose (ID-183: 4 UI/mL; ID-5 and ID-244: 2 UI/mL). After the fourth dose, two of them showed anti-HBs of >10 UI/mL (ID-183: 109.60 UI/mL; ID-5: 13.9 UI/mL)

## 4. Discussion

In this study, we showed there are gaps in the achievement of the Millennium Development Goal to eliminate viral hepatitis by 2030 in Brazil. Notably, this investigation was conducted before the COVID-19 pandemic, which disrupted core services for the prevention, control, and treatment of viral hepatitis [14].

The high proportion of anti-HAV positivity was predictable. However, in the last decades, Brazil’s HAV endemicity profile changed to intermediate [19], and the finding that 36% of individuals aged ≤21 years are HAV-susceptible is concerning. Unfortunately, data on HAV among TGW in Brazil are scarce. Recently, in another city in Central Brazil, Castro et al. [20] reported an overall anti-HAV prevalence of 69.7%, ranging from 62.3% among MSM (*n* = 276) to 83.2% among TGW (*n* = 149). Among MSM aged ≤20 years, 60% were HAV-susceptible. In addition, Castro et al. [20] observed one TGW who was anti-HAV-IgM-positive and, thus, a potential HAV disseminator in her network. She was aged 21 years, reported sex work, alcohol, and drug use, and did not use a condom during oral sex. In addition, HAV outbreaks among MSM have been reported in low-HAV-endemicity regions [21,22,23], and an increased number of hepatitis A cases due to sexual transmission has been observed in Brazil [24,25]. Thus, more studies on HAV epidemiology are necessary to evaluate the impact of Brazilian social and economic advances on HAV prevalence among higher-risk individuals, such as young TGW and MSM, and the inclusion of the hepatitis A vaccine in the National Immunization Program for these populations.

Brazil has low endemicity for hepatitis B and C [19,26]. However, hepatitis B and C are still poorly studied among TGW in Brazil. The prevalence of anti-HCV in this study was almost 2.5-fold higher than that estimated by Benzaken et al. [26] (0.53%) in the general Brazilian population. Although injection-drug use is a significant risk factor in HCV infection [14,27,28], it was reported with a low frequency in this study (2.7%). However, the high frequency of people living with HIV and risk behaviors (unsafe sex, non-injection drug use, and sex work) among TGW may also put them at risk of acquiring HCV and likely keeping the virus circulating in the TGW social network. Moreover, all the anti-HCV-positive individuals were unaware of their HCV serological status, confirming a gap in hepatitis C diagnosis and the importance of a population-based approach to meeting the global target of the Sustainable Development Goals and the global health-sector strategy [14]. Therefore, HCV surveillance in critical populations should be a priority, especially in the era of pre-exposure prophylaxis for HIV, which has been associated with relaxing safe-sex standards [29].

In this study, 12.3% of TGW were exposed to HBV, and 0.46% were still infected, ratifying the assumption of low HBV endemicity. However, this prevalence was almost six times higher than that found in blood donors in Central Brazil (1.9%) [30]. Furthermore, only 41.5% (95% CI, 36.94–46.12) of TGW showed a serological profile of previous HBV vaccination, and this was similar to that shown in a study in the same region with 522 MSM (40.3%; 95% CI, 32.3–48.8) that included TGW, suggesting low hepatitis B vaccination coverage in this population in Central Brazil [12]. This was consistent with that reported in other countries. For example, Luzzatti et al. [31] reported 38.5% of HBV immunity among 218 TGW who were referred to an Italian center for total-sex-reassignment surgery. In comparison, Adeyemi et al. [13] found 5% HBV vaccination among 717 Nigerian MSM and TGW.

While the proportion of TGW vaccinated against HBV did not differ among the age groups, an increasing proportion of HBV exposure was found according to age. Furthermore, this variable remained associated with HBV exposure after adjusting factors such as sex work, number of sexual partners, and living with HIV. In Brazil, hepatitis B vaccination has been available free of charge for infants since 2001 and gradually expanded to older individuals. Thus, the much younger TGW should have been vaccinated before sexual initiation and protected from acquiring HBV despite their multiple risky behaviors. Our findings ratify this assumption.

The standard scheme of the hepatitis B vaccine is a fundamental cause of non-adherence to the three vaccine doses among high-risk populations, mainly owing to long intervals between second and third vaccine doses [15]. Therefore, some authors have suggested shorter schemes to overcome this difficulty [15,16,17]. There was better adherence with three doses using a super-accelerated scheme (G1) than with a conventional scheme (G2). The accelerated scheme increased compliance by 14 percentage points. Furthermore, although the GMT for anti-HB titers was higher in G2 vs. G1, it overlapped between the groups, indicating no immunogenicity difference between the schemes.

Furthermore, in G1, the GMT for the anti-HB titers was similar after the third and fourth doses. Two individuals who did not develop protective anti-HB titers after the third vaccine dose developed them after the fourth dose. Thus, where possible, a fourth dose should be offered.

This study has some limitations. First, the data were collected in Goiás, Central Brazil, and may not represent the general Brazilian TGW population, although their sociodemographic characteristics are similar to those reported in another Brazilian study [9]. This study used a crude estimator for RDS sampling instead of the RDS II estimator because the actual performance of this method remains largely unknown [32]. Therefore, overestimation may not be ruled out, although the sample size (*n* = 440) should have minimized it. Finally, the anti-HB marker alone was used to indicate previous HBV vaccination. Anti-HB titers may decline over time, although the protection against hepatitis B is maintained [33]. Therefore, some TGW may have been vaccinated previously and mischaracterized as HBV-vulnerable. However, in the absence of a vaccination card (the gold standard), this is the most appropriate indicator of previous hepatitis B vaccination. Finally, these women have high geographic mobility, contributing to the increased number of losses and their difficulties with in complying with the time intervals between vaccine doses.

## 5. Conclusions

The finding that 36% of TGW aged ≤21 years are HAV-susceptible deserves public-health attention. The higher anti-HCV prevalence among TGW compared to that in the general Brazilian population and the detection of HBsAg positivity among TGW suggest that risky behaviors may keep these viruses circulating in this vulnerable population in Central Brazil. Additionally, the low prevalence of serological evidence of HBV immunization may also contribute to viral dissemination. The results of better compliance with the super-accelerated HBV vaccine schedule and good immunogenicity support a shorter vaccination schedule for TGW, a hard-to-reach population.

## Figures and Tables

**Figure 1 tropicalmed-07-00269-f001:**
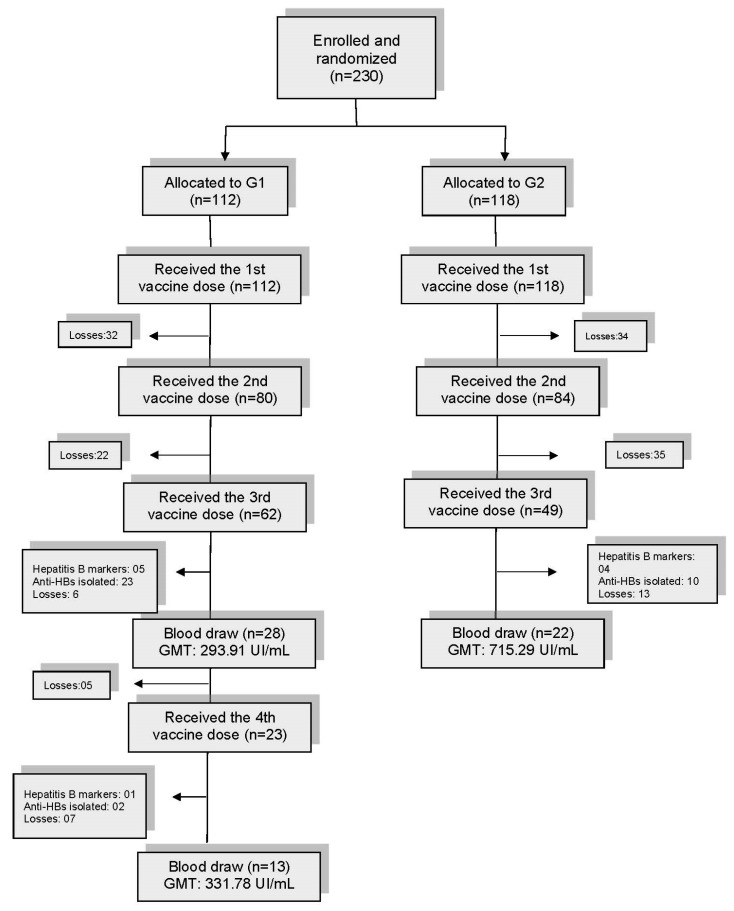
Flowchart of hepatitis B vaccination.

**Table 1 tropicalmed-07-00269-t001:** Characteristics of 440 transgender women in Central Brazil.

Sociodemographic Details	*n* ^a^	%
Age group (median 25; IQR: 9)		
Monthly income (US$) * (median 511.5; IQR: 511.5)		
Formal education *		
Full or partial undergraduate level	48	10.91
Full or partial secondary level	270	61.36
Until fundamental level	122	27.76
Color/race (self-defined) (NI:2)		
White	90	20.55
Black or Brown	312	71.23
Indian or Asiatic	36	8.22
Marital status		
Single/divorced/widow	375	85.42
Married (official or consensual)	64	14.57
Sexual		
Number of sexual partners in the month (median: 100; IQR:396)		
Exchange sex for money, drugs, and goods		
No	47	10.68
Yes	393	89.32
History of STI		
No	220	50.0
Yes	220	50.0
Condom in last sexual intercourse		
No	293	67.98
Yes	138	32.02
Oro-anal sex		
No	229	52.16
Yes	210	47.84
Insertive anal sex		
No	113	25.74
Yes	326	74.26
Receptive anal sex		
No	11	2.51
Yes	428	97.49
Previous positive HIV testing		
No	305	69.32
Yes	77	17.5
No testing	58	13.18
Parenteral		
Illicit-injection-drug use		
No	425	97.25
Yes	12	2.74
Illicit-non-injection-drug use		
No	114	25.97
Yes	325	74.03
Industrial silicone application		
No	242	55.12
Yes	197	44.87
Daily alcohol use		
No	377	89.67
Yes	58	13.33

^a^ Only valid data; * During the study period, USD 1 was equivalent to BRL 3.91. HIV, human immunodeficiency virus; IQR, interquartile range.

**Table 2 tropicalmed-07-00269-t002:** Prevalence of viral hepatitis serological markers among transgender women.

Marker	*n*	Pos.	%	95% CI
Anti-HAV IgM	439	01	0.23	0.04–1.28
Anti-HAV total	439	332	75.63	71.4–79.71
Anti-HCV	439	06	1.37	0.63–2.95
HBsAg + anti-HBc total	439	02	0.46	0.12–1.65
Anti-HBs + anti-HBc total	439	37	8.43	6.18–11.4
Anti-HBc only	439	15	3.42	2.08–5.56
Any HBV infection marker	439	54	12.3	9.55–15.7
Anti-HBs only	439	182	41.5	36.94–46.12

CI, confidence interval; HAV, hepatitis A virus; HBc, hepatitis B core antigen; HCV, hepatitis C virus; HBsAg, hepatitis B surface antigen; HBV, hepatitis B virus; Pos., positive.

**Table 3 tropicalmed-07-00269-t003:** Multivariate analysis of factors associated with HBV exposure (anti-HBc) among transgender women in Central Brazil.

Variable	Anti-HBc Marker				
	Pos./Total	%	*p*	OR (CI 95%)	AdjOR (CI 95%)
Age (years)	5/72	6.9		1.00	
≤21	7/70	10.0	0.532	1.49 (0.45–4.93)	1.61 (0.48–5.38)
22–25	11/56	19.6	0.037	3.28 (1.07–10.06)	3.38 (1.08–10.54)
26–30	31/59	52.5	<0.001	14.84 (5.23–42.07)	13.33 (4.44–40.0)
>30					
Education					
Above fundamental level	37/177	20.9		1.00	
Up to fundamental level	17/80	21.3	0.950	1.02 (0.53–1.95)	
Monthly income (US$)					
<500	23/140	16.4			
≥500	31/117	26.5	0.049	1.83 (1.00–3.36)	
White color					
Yes	12/49	24.5			
No	41/207	19.8	0.467	0.76 (0.36–1.59)	
Single					
No	10/36	27.8			
Yes	44/221	19.9	0.283	0.65 (0.29–1.44)	
Sex work					
No	33/99	33.3			
Yes	21/158	13.3	<0.001	0.31 (0.16–0.57)	
History of STI					
No	25/137	18.2			
Yes	29/120	24.2	0.245	1.43 (0.78–2.61)	
Condom in the last sexual intercourse					
Yes	35/176	19.9			
No	19/76	25.0	0.364	1.34 (0.71–2.54)	
Sexual intercourse with partner STI carrier					
No	36/172	20.9			
Yes	12/52	23.1	0.741	1.13 (0.54–2.38)	
Group sex					
No	11/45	24.4			
Yes	43/212	20.3	0.534	0.79 (0.37–1.68)	
Insertive anal sex					
No	15/61	24.6			
Yes	39/196	19.9	0.432	0.76 (0.39–1.50)	
Receptive anal sex					
No	2/7	28.6			
Yes	52/250	20.8	0.619	0.66 (0.12–3.48)	
Previous positive HIV testing					
No	33/177	18.6			
Yes/no testing	21/80	26.3	0.166	1.55 (0.83–2.90)	
Non-injection illicit-drug use					
No	15/63	23.8			
Yes	38/193	19.7	0.483	0.78 (0.40–1.55)	
Injection illicit-drug use					
No	53/249	21.3			
Yes	0/5	-	0.246		
Daily alcohol use					
No	46/213	21.6			
Yes	6/41	14.6	0.312	0.62 (0.25–1.57)	

OR, odds ratio; AdjOR (95% CI), Adjusted odds radio (95% confidence interval); HBc, hepatitis B core antigen; HIV, human immunodeficiency virus; Pos., positive; STI, sexually transmitted infection. Variables included in multiple regression model were age, monthly income, sex work, and previous positive HIV testing.

## Data Availability

The data presented in this study are available on request from the corresponding author.

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
