# Peer review of "Viral Hepatitis A, B and C in a Group of Transgender Women in Central Brazil"

_tropicalmed, 2022, doi:10.3390/tropicalmed7100269_

Round 1
Reviewer 1 Report
The authors describe the prevalence of viral hepatitis markers in a population of TGW in Brazil. The study is well designed and the methodology and results are well presented. The discussion is well written and compares the findings of the study to the literature data on previous studies and the prevalence of the general population.
Author Response
Dear Reviewer,
Thank you for your careful review.
Best regards,
Sheila A. Teles
Reviewer 2 Report
The study is well designed and described and takes into account the possible limitations of the study.
There would be only one small issue to clarify:
There are a significant number of volunteers who test positive for HIV. Although it is not the objective of the study it would be important to know the clinical/immunological status of these volunteers. Although it would not be a strong bias, volunteers who may be immunocompromised could underestimate the total seroprevalence of each of the viral hepatitis tested. Loss of antibodies has been described in some clinical contexts compatible with immunosuppression/immunosuppression.
Author Response
Dear Reviewer:
Thank you for the suggestion about our manuscript. Unfortunately, we don't have the clinical/immunological status of the participants. However, it is noteworthy they were young and apparently healthy.
Sincerely, Sheila A. Teles
Reviewer 3 Report
In the submitted manuscript, Ferri et al. investigated the incidence of different viral hepatitis in a group of transgender Brazilian women. In addition, immunological characterization of HBV vaccination has been presnted. The study is interesting and well-written.
Author Response
Dear Reviewer,
Thanks for the careful review.
Introduction and Reference: We included a systematic review and metanalysis recently publicized (2022) in the Introduction.
In conclusion, we reinforce the importance of HBV vaccination.
We hope these alterations are following your recommendation.
Best regards,
Sheila